# Neoadjuvant Intratumoral Immunotherapy with Cowpea Mosaic Virus Induces Local and Systemic Antitumor Efficacy in Canine Mammary Cancer Patients

**DOI:** 10.3390/cells12182241

**Published:** 2023-09-08

**Authors:** Guillermo Valdivia, Daniel Alonso-Miguel, Maria Dolores Perez-Alenza, Anna Barbara Emilia Zimmermann, Evelien Schaafsma, Fred W. Kolling, Lucia Barreno, Angela Alonso-Diez, Veronique Beiss, Jessica Fernanda Affonso de Oliveira, María Suárez-Redondo, Steven Fiering, Nicole F. Steinmetz, Johannes vom Berg, Laura Peña, Hugo Arias-Pulido

**Affiliations:** 1Department of Animal Medicine, Surgery and Pathology, Mammary Oncology Unit, Veterinary Teaching Hospital, Veterinary Medicine School, Complutense University of Madrid, 28040 Madrid, Spain; edgargva@ucm.es (G.V.); danialon@ucm.es (D.A.-M.); mdpa@ucm.es (M.D.P.-A.); lbarreno@ucm.es (L.B.); angalo02@ucm.es (A.A.-D.); marsuare@ucm.es (M.S.-R.); laurape@ucm.es (L.P.); 2Institute of Laboratory Animal Science, University of Zurich, 8952 Schlieren, Switzerland; annabarbaraemilia.zimmermann@uzh.ch (A.B.E.Z.); johannes.vomberg@uzh.ch (J.v.B.); 3Aquila Data Analytics, LLC., Concord, NH 03766, USA; aquiladataanalytics@gmail.com; 4Dartmouth Cancer Center, Geisel School of Medicine at Dartmouth, Lebanon, NH 03756, USAsteven.n.fiering@dartmouth.edu (S.F.); 5Department of NanoEngineering, University of California San Diego, 9500 Gilman Dr., La Jolla, CA 92093, USA; verobeiss@googlemail.com (V.B.); jaffonsodeoliveira@eng.ucsd.edu (J.F.A.d.O.); nsteinmetz@ucsd.edu (N.F.S.); 6Department of Microbiology and Immunology, Geisel School of Medicine at Dartmouth, Lebanon, NH 03756, USA; 7Department of Radiology, University of California San Diego, La Jolla, CA 92093, USA; 8Department of Bioengineering, University of California San Diego, La Jolla, CA 92039, USA; 9Moores Cancer Center, University of California San Diego, La Jolla, CA 92039, USA; 10Center for Nano Immuno-Engineering, University of California San Diego, La Jolla, CA 92039, USA; 11Institute for Materials Discovery and Design, University of California San Diego, La Jolla, CA 92039, USA; 12Center for Engineering in Cancer, Institute for Engineering in Medicine, University of California San Diego, La Jolla, CA 92039, USA

**Keywords:** canine mammary cancer, intratumoral immunotherapy, plant virus, cowpea mosaic virus, nanoparticles, immune cells, tumor microenvironment, patient outcome

## Abstract

The lack of optimal models to evaluate novel agents is delaying the development of effective immunotherapies against human breast cancer (BC). In this prospective open label study, we applied neoadjuvant intratumoral immunotherapy with empty cowpea mosaic virus-like particles (eCPMV) to 11 companion dogs diagnosed with canine mammary cancer (CMC), a spontaneous tumor resembling human BC. We found that two neoadjuvant intratumoral eCPMV injections resulted in tumor reduction in injected tumors in all patients and in noninjected tumors located in the ipsilateral and contralateral mammary chains of injected dogs. Tumor reduction was independent of clinical stage, tumor size, histopathologic grade, and tumor molecular subtype. RNA-seq-based analysis of injected tumors indicated a decrease in DNA replication activity and an increase in activated dendritic cell infiltration in the tumor microenvironment. Immunohistochemistry analysis demonstrated significant intratumoral increases in neutrophils, T and B lymphocytes, and plasma cells. eCPMV intratumoral immunotherapy demonstrated antitumor efficacy without any adverse effects. This novel immunotherapy has the potential for improving outcomes for human BC patients.

## 1. Introduction

Early breast cancer (BC) detection and therapy have decreased BC-related death rates by ~38% [1], but BC remains the leading malignancy in women in the United States, with ~298,000 new cases and ~43,000 BC-related deaths being expected in 2023 [2]. Commonly used chemotherapy-based treatments are not efficacious and have considerable life-altering side effects; most BC patients who develop metastatic disease will succumb to it [3,4]. These sobering statistics highlight the urgent need for innovative therapies that reduce tumor burden, prevent, or eliminate metastasis, and improve survival and quality of life in BC patients. Although anthracycline-taxane-based chemotherapy, combined with endocrine therapy and targeted therapy, when appropriate, remains the backbone of neoadjuvant therapy for BC [5,6,7], its efficacy is minimally effective as shown by a dismal 5-year survival rate of 29% for BC patients with distant disease [8]. Furthermore, while anti-programmed cell death 1 (PD-1) immunotherapy is now approved for early high-risk and advanced triple-negative (TN) BC, only ~20% of patients benefit from this therapy [9,10,11]. The inability to identify effective therapies has multiple factors, but a major contributing factor is the absence of optimal models to test new therapies outside of mice, which do not accurately model breast cancer [12].

Companion dogs with spontaneous mammary cancer are a valuable animal population to test and study new therapeutic approaches before human clinical trials. Canine mammary cancers (CMCs) share clinicopathological, genomic and immune features with human BC [13,14,15,16,17]. Furthermore, canine cancer patients are outbred animals, have intact immune systems and tumors that, like spontaneous human tumors, are predominantly “self” immunologically, making them a uniquely valuable resource to evaluate the clinical efficacy of new anti-cancer agents, approaches or combinations, particularly immunotherapy [13,14,15,16]. CMC is rare in the USA because female dogs are generally ovariectomized early in life, which drastically reduces its prevalence. However, in countries where early ovariectomy is not usually performed, CMC is the most frequent neoplasia in intact female dogs, accounting for almost 53% of all canine neoplasms [18]. In Spain, most female dogs are not spayed early in life, and therefore, the number of CMC patients is high [19,20]. The standard of care for CMC patients is surgical intervention within 2–3 weeks after initial cancer diagnosis followed by post-surgical adjuvant medical therapy for high-risk CMC patients [14].

We have extensively documented the ability of intratumoral immunotherapy using cowpea mosaic virus (CPMV) nanoparticles and empty CPMV (eCPMV) virus-like nanoparticles to stimulate antitumor immune responses and improve outcomes in various syngeneic murine tumor models, including breast cancer [21,22,23], companion dogs with canine oral melanoma [24] and canine inflammatory mammary cancer (CIMC) [25]. CPMV acts by delivering strong immunostimulatory signals through multiple Toll-like receptors (TLRs 2, 4, and 7) that change the tumor microenvironment (TME) from immune suppressive to immune stimulatory, and generate local antitumor immunity that develops into systemic antitumor immunity which opposes metastatic disease [26]. CPMV and eCPMV are identical in their protein content and both are highly immunogenic through TLR stimulation; however, eCPMV lacks viral RNA [21,27].

In this study, we evaluated the clinical efficacy of immunomodulatory eCPMV nanoparticles against CMC. Our results demonstrated robust clinical efficacy of neoadjuvant eCPMV intratumoral immunotherapy (eCPMV immunotherapy from here on), leading to tumor reduction in both injected tumors and frequently in noninjected tumors present in the ipsilateral and contrallateral mammary chains in treated CMC dogs. The therapy was safe, and the response was observed in all dogs independent of clinical stage, tumor size, histological tumor grade or tumor molecular subtype. This study supports the implementation of this novel neoadjuvant immunotherapy for human BC patients, for whom there are no efficacious immunotherapies.

## 2. Materials and Methods

### 2.1. Canine Patient Recruitment and Selection Criteria

This prospective open label study was performed at the Mammary Oncology Unit of the Veterinary Teaching Hospital of the Complutense University of Madrid, Madrid, Spain, from October 2018 to February 2022, when the last CMC patient was enrolled. All patients and their guardians signed an informed consent. This study was approved by the Animal Experiment Ethics Committee of the Veterinary Teaching Hospital of Complutense University (Study #04/2018). The inclusion criteria at diagnosis are described in Appendix A. The characteristics of 11 individual eCPMV-treated dogs are described in Table 1 and Appendix A. The clinical staging system, histopathological classification of tumors and the histological grade of malignancy were evaluated as described elsewhere [28,29,30].

### 2.2. eCPMV Immunotherapy Treatment Protocol

eCPMV nanoparticles were produced in plants as described previously [31]. The largest tumor in each patient was the target tumor (*injected*) for eCPMV immunotherapy. Mammary nodules (malignant and benign tumors) present in the same and contralateral chains were observed to evaluate the systemic impact of eCPMV on *noninjected* nodules in the same canine patient (Appendix A). In the enrolled patients, a pretreatment incisional biopsy of the target-injected tumor was taken before the eCPMV immunotherapy, followed by an intratumoral eCPMV injection immediately after obtaining the tumor biopsy sample. Two intratumoral doses of eCPMV immunotherapy were administered, the first on day 0 (D0) and the second between days six and nine (D6–D9; DTx2), followed by surgery on D12-D17 (DSx) at which time the injected tumor was resected, and a surgical biopsy was collected for histopathology and research studies. Surgical procedures were performed per institutional standard of care protocol (described in Appendix A). The eCPMV nanoparticles (0.2 mg per injection) were diluted in 0.5 mL of sterile phosphate-buffered saline (PBS) and injected using a 25G needle. The injected PBS volume was equally distributed in three to five locations within a treated tumor. Doses were based on previously published studies in mice [21,22,23] and dogs [24]. The eCPMV immunotherapy was provided as a neoadjuvant therapy and after surgery, adjuvant therapy was provided. This adjuvant therapy was maintained for two years unless local recurrence or metastases were observed. After surgery, follow-up was performed every three months until two years or until death or euthanasia due to tumor progression or any other cause. Thoracic radiographs and abdominal ultrasound were performed every three months to search for metastases.

### 2.3. Quality of Life (QOL) and Tumor Response Evaluation

Each canine patient was closely observed by the attending veterinarian in the clinic for 4 h after each intratumoral eCPMV injection and subsequently daily by owners to evaluate the potential risk induced by eCPMV immunotherapy. The QOL of each patient was evaluated before eCPMV immunotherapy, before the second eCPMV injection, and at surgery day using a preestablished survey [32].

The tumor response to eCPMV immunotherapy was evaluated once a week during the treatment period by measuring the tumor volume (Tv) using the formula Tv = 0.5 × long axis × (short axis)^2^. The eCPMV-injected tumor and any noninjected mammary nodules in the same or contralateral chains were evaluated in a similar manner. The percentage of tumor growth inhibition (%TGI) was estimated as %TGI = 100 × (final Tv − initial Tv)/initial Tv. All measurements are in cubic centimeters (cm^3^). Although the Response Evaluation Criteria in Solid Tumors version 1.1 (RECIST 1.1) and guidelines for immunotherapeutic trials (iRECIST) are unsuitable for intratumoral immunotherapy trials because they were designed for systemic therapy [33], we applied the itRECIST criteria for exploratory analyses [33].

### 2.4. Hematological, Biochemical, Flow Cytometry, and Cytokine Analyses

Systemic changes induced by eCPMV immunotherapy were evaluated using a blood sample (~10 mL) collected from each patient at D0, DTx2, DSx, and 30 days after surgery (D45) for hematologic, biochemistry, cytokine, and peripheral blood mononuclear cells (PBMCs) analyses as previously described [25] (detailed in Appendix A). The evaluation of hematological, biochemical, and other adverse events related to eCPMV immunotherapy was conducted per the Veterinary Cooperative Oncology Group criteria (Version 2) [34].

### 2.5. Detection of Anti-eCPMV Antibodies in Canine Plasma

Enzyme-Linked Immunosorbent Assay (ELISA) was used to detect levels of eCMPV-specific IgG (immunoglobulin G) titers or anti-drug antibodies (ADAs) on plasma samples collected at various times during the trial as described elsewhere [35] (detailed in Appendix A).

### 2.6. Histopathology and Immunohistochemistry (IHC) Assays

Single 3 mm tumor tissue sections were used for histopathology and IHC assays for Ki67, estrogen receptor, progesterone receptor, human epidermal factor receptor-2 (HER2), myeloperoxidase, CD3, FoxP3, CD20, and MUM1. IHC details and scoring of markers are described in Appendix A.

### 2.7. RNA Analyses

Snap-frozen tumor samples were used for RNA isolation and RNA-seq analysis; the details of the RNA-seq and bioinformatic analyses are provided in Appendix A.

### 2.8. Statistical Analyses

Primary outcomes were efficacy, measured by the reduction in tumor volume of the injected target tumor and tumor reduction in noninjected mammary nodules; and biosafety, evaluated through the analysis of hematological and biochemistry changes in blood. Paired Student’s *t*-test or Wilcoxon test were used as appropriate. Two-tailed *p* values less than 0.05 were considered statistically significant. Full details of statistical analysis are provided in the Appendix A.

## 3. Results

### 3.1. Epidemiological and Clinico-Pathological Characteristics of CMC Patients

The CMC population had tumors representing the same spectrum observed in human BC: mostly old-age patients of different breeds and weights, various tumor subtypes (luminal A and B, and TN), low (I), intermediate (II) and high histological grade (III) tumors, and clinical stage I to IV tumors (Table 1 and Appendix A). In addition to the target injected tumor, the number and location of mammary nodules varied in eCPMV-treated dogs (Appendix A, and Table 2 and Table 3). Some dogs had nodules only in the ipsilateral chain (P2, P8, and P10), the contralateral chain (P3 and P4) or in both mammary chains (P1, P5, P7, P9, and P11) (Table 3). P6 was the only CMC patient with one single tumor mass treated with eCPMV (Table 2).

### 3.2. eCPMV Immunotherapy Induces Tumor Reduction in Injected and Noninjected CMC Tumors

In the 11 treated dogs, eCPMV immunotherapy induced a measurable tumor reduction in all eCPMV-injected tumors (Figure 1A). The first eCPMV injection led to 3.8% to 40.3% reduction in %TGI rates, and a further tumor reduction was observed on the day of surgery after the second eCPMV injection when %TGI rates varied from 5.8% to 63.2% (Figure 1A and Table 2). Of note, tumor growth was observed in P8’s largest tumor after the first injection at D7, with a subsequent tumor reduction at surgery day (D12) (Figure 1A and Table 2). Because the tumor size at D12 was less than at D0, tumor growth at D7 is considered pseudoprogression.

Notably, eCPMV injections induced tumor reduction in noninjected tumors located in the ipsilateral (Figure 1B) and contralateral (Figure 1C,D) mammary chains of the same eCPMV-treated dog. The tumor reduction varied between CMC patients and individual lesions with %TGI rates fluctuating in a wide range with no clear correlation to tumor size, histopathology grade, or chain location (Table 3). Eight dogs with grade I carcinomas (P1-P5 and P7-P9), one with grade II (P11), and two with grade III carcinomas (P6 and P10) received eCPMV in the target tumor. In these dogs, tumor reduction was observed in all noninjected tumor grade I carcinomas present in the ipsilateral chain, including two large tumors in P1 and P8 (1.7 cm^3^ and 1.3 cm^3^, respectively); P7 also had a grade II carcinoma responsive to eCPMV immunotherapy. P11, a grade II carcinoma, was the only dog without tumor reduction in one grade I carcinoma and two grade II carcinomas present in the ipsilateral chain (Table 3). The tumor reduction was also observed in dysplasias present in the ipsilateral chain in P1 (two large masses) and P7 (one small mass), but not in P9 which had a small mass. The tumor reduction varied in the benign tumors with response in P2, P7 and P8, including a large mass (2.5 cm^3^) in P8, and no response observed in the P2 and P9 nodules (Table 3).

The tumor reduction varied in the contralateral mammary chain with no effect on grade I carcinomas (P1 had no tumor growth, and P7 had tumor growth in two tumors); had a positive effect in one dysplasia (P1); induced a response in some benign tumors (P3 and P9, one mass each); and had no effect in other benign tumors (P1 and P3, one mass each, and P4, two nodules). Of note, P5 had one and P11 had eight tumor masses (unresected, all of unknown histopathologic diagnosis), with tumor reduction occurring in one (P5) and four (P11) nodules. Also, P11 presented one nodule without size change, tumor growth in two nodules, and one nodule had pseudoprogression, followed by tumor reduction to the original size (Table 3).

Following the itRECIST criteria, the response to eCPMV immunotherapy was stable disease (SD) in target injected tumors (Appendix A), SD and partial response (PR) in target noninjected lesions (Appendix A), and present in nontarget noninjected lesions (defined as the tumor that neither disappeared nor had an unequivocal progression) (Appendix A); the classification of each itRECIST criteria is described in Appendix A.

### 3.3. eCPMV Immunotherapy Is not Toxic

The therapy did not cause significant adverse events. No adverse reactions were observed systemically or at the injection site during the 4 h observation period after each eCPMV administration. eCPMV immunotherapy did not induce significant fluctuations in hematocrit and hemoglobin levels in any dog during the treatment period. Fluctuations remained within the normal range in most eCPMV-treated dogs with a significant increase in hemoglobin levels observed at D45 (*p* = 0.033; Appendix A). Furthermore, there were no significant changes in glucose, urea, creatinine, and ALT during eCPMV immunotherapy. A significant increase in total serum proteins was observed after the first eCPMV injection in some dogs, followed by a steady decrease to normal levels by surgery day (*p* = 0.044; Appendix A). These fluctuations did not trigger any medical intervention, suggesting that eCPMV immunotherapy with this dosing does not negatively affect hepatic, renal and digestive functions in treated dogs.

### 3.4. eCPMV Immunotherapy Induces Modest Changes in Immune Blood Cell Populations

Peripheral white blood cell analysis shows non-significant fluctuations in lymphocyte and monocyte numbers during eCPMV immunotherapy; and a significant increase in mature and immature neutrophil numbers after the first eCPMV injection (*p* < 0.05 for both), with a subsequent decrease close to pre-treatment levels (Appendix A and Appendix A). These findings suggest that eCPMV immunotherapy induces a transient systemic increase in peripheral blood inflammatory cells. Flow cytometry on canine PBMC demonstrated that eCPMV induced limited differential fluctuations in various immune cells (gating strategy is illustrated in Appendix A): during treatment, there was an increase in CD8^+^granzyme B (GZMB)^+^ T cells, CD4^+^ cells, and T regulatory (Treg) cells, and a steady decrease in T cells, CD8^+^ T cells, NK cells, B cells, and APC cells, and monocytes; the Treg^+^/CD8^+^ T cell ratio in peripheral blood was not affected (Appendix A). Nonsignificant fluctuations were observed in three monocytes subsets identified in canine monocyte nomenclature as major histocompatibility complex II (MHCII)^+^CD4^+^ (Mo1), MHCII^−^CD4^+^ (Mo2), and MHCII^+^CD4^−^ (Mo3): Mo1 slightly increased, and Mo2 and Mo3 decreased during treatment, and both Mo1 and Mo2 decreased, while Mo3 increased by D45 (Appendix A).

### 3.5. eCPMV Immunotherapy Induces Changes in Cytokine Plasma Levels

Of the 13 cytokines analyzed, a significant decrease in interleukin (IL)-6, IL-7, and monocyte chemoattractant protein-1 (MCP-1) levels were observed after the first eCPMV injection (*p* < 0.04 for all), followed by an increase close to basal plasma levels by surgery day; a nonsignificant decline in IL-2 plasma levels was observed, and a marginally significant decrease in IL-10 by DSx (*p* = 0.050) (Appendix A) was also observed. Average changes in cytokine plasma levels in all eCPMV-treated patients during treatment are shown in Appendix A, and individual changes are shown in Appendix A. These data indicate that eCPMV immunotherapy does not induce large systemic release of cytokines which could lead to serious adverse effects.

### 3.6. Anti-CPMV Antibodies Do Not Block Treatment Efficacy

The presence of antibodies against CPMV nanoparticles was assayed in five CMC patient’s serum before, during, and after therapy (Appendix A). Prior to treatment, all five dogs had detectable anti-CPMV antibodies. The first eCPMV injection led to a ~three-fold increase in anti-CPMV antibodies in the plasma of P1, P2, and P4, slight or no increase in P3 and P5 (which were already quite high), and the levels remained high beyond 200 days after the first eCPMV injection in P1 and P2 (Appendix A). Tumor growth inhibition was observed in all injected tumors as well as in many noninjected tumors and hyperplasia-dysplasia, and in benign masses, while no responses were observed in some tumors and benign masses (Figure 1, Table 2 and Table 3). Of note, in patients P3 and P5, where anti-CPMV antibodies were high prior to treatment, P3 had a 78% TGI only in its ipsilateral grade I tumor, and P5 had two contralateral benign masses with a 59% TGI responses observed in the largest one (0.79 cm^3^) and no response in the smallest one (0.07 cm^3^) (Table 3). Given the observed wide range of tumor reduction in both injected and noninjected masses in most of the dogs, these data imply that anti-CPMV antibodies do not inhibit eCPMV immunotherapy, as was previously demonstrated in mice with ovarian cancer [35].

### 3.7. eCPMV Immunotherapy Induces Significant Changes in the TME

To gain more insights into the potential changes in the TME induced by eCPMV, bulk RNA-seq was performed on pre-treatment (D0) and post-treatment (D12-D17) tumor biopsies. Principal component analysis (PCA) indicated that the predominant axis of variation (Principal Component 1) was significantly associated with treatment (Figure 2A and Appendix A, *p* = 0.04). This suggests that modulation of the TME by eCPMV immunotherapy can be observed despite high heterogeneity among patients (tumor size, clinical stage, breed, age, etc.). Differential gene expression analysis revealed a number of differentially expressed genes (DEGs) when comparing post- to pretreatment samples (false discovery rate (FDR) < 0.1, up = 16 genes, down = 1 gene) (Figure 2B). Among the top upregulated genes was GAS1, a gene that plays a role in growth suppression by blocking the entry to S phase and prevents cycling of normal and transformed cells. Other genes included ANK2 and SYNPO2, both involved in actin rearrangements. These changes suggest alterations in cell cycle and cytoskeletal processes (Appendix A). Pathway analysis further supported the downregulation of DNA replication pathways and the upregulation of extracellular fiber rearrangements (Figure 2C). In addition, several transcription factor genes were significantly upregulated (Appendix A), as well as immune-related pathways involved in CD4+ T cell activation (Appendix A).

As eCPMV immunotherapy stimulates antitumor immune responses [21,22,23], we used RNAseq to infer the abundance of several immune cell types known to infiltrate the TME. We found slight changes in the immune infiltration level. For example, compared to pretreatment samples, activated dendritic cells (DCs) trended to be increased in posttreatment samples (Figure 2D, *p* = 0.06), whereas M1 macrophages decreased upon eCPMV immunotherapy (Figure 2E, *p* = 0.09).

Additional changes in immune cell infiltration were detected via IHC analysis. A significant increase in the numbers of neutrophils, T lymphocytes, B lymphocytes, Treg lymphocytes, and plasma cells was observed in posttreatment tumor biopsies when compared to pretreatment tumor biopsies (*p* < 0.02 for all; Figure 2F–H,J and Appendix A). Although a significant increase in the number of Treg lymphocytes was observed (*p* = 0.016; Figure 2I), the Treg/Total T cell ratio (FoxP3^+^/CD3^+^) significantly decreased (*p* = 0.003; Figure 2K). eCPMV immunotherapy did not induce significant changes in Ki67 proliferation index (Appendix A). These findings highlight the strong immunogenicity of eCPMV nanoparticles which induced a significant increase in innate and adaptive immune cell populations within the TME in eCPMV-treated CMC patients.

### 3.8. eCPMV Immunotherapy Does Not Affect QOL and Is Associated with Improved Survival in CMC Patients

During the observation time after the first eCPMV injection, QOL was improved in one dog (P1), no changes were observed in nine dogs (P2 to P10), and the condition worsened in one dog (P11; per owner’s report, it was due to skin suture-related itching).

Of the 11 enrolled CMC patients, 1 eCPMV-treated patient died of non-cancer-related events, and the remaining 10 eCPMV-treated remained alive as of 19 July 2022 when the survival information was updated (Table 1). Although the study was not powered to perform survival analysis, the mean survival time for all eCPMV-treated patients was 758 days (standard deviation, 461), for histopathologic grade I (8 out of 11 patients; all alive), 930 days (SD, 420), grade II, 267 days (1 patient; alive), and grade III, 316 days (SD, 139; 2 patients, 1 alive and the other died of non-cancer-related event); no recurrences and/or metastasis have been documented. As a reference, a two-year follow-up of 65 CMC patients who underwent only surgery at the same institution found that the mean survival time was ~1160 days (100% alive), 976 days (84% alive), and 610 days (41% alive) for histological grade I, II, and III CMC patients, respectively. Of note, the presence of recurrences and/or metastases was observed in 1 of 29 (3%), 3 of 19 (17%), and 10 of 17 (59%) patients with grade I, II, and III, respectively [29].

## 4. Discussion

Although anti-PD-1 immunotherapy has been useful in treating some BC patients, it is only approved for early high-risk and advanced TN BC patients; only a minority of these patients benefit from this therapy [9,10,11]. Identifying immune agents that couple broad efficacy across all BC subtypes while limiting host toxicity has been challenging. This is due, in part, to the lack of optimal models which can provide reliable translational information. Given the shared clinical and pathophysiological presentation, and genomic and immune features between CMC and human BC patients [13,14,15,16,17], we explored whether eCPMV, a potent immunogenic agent [21,36], could provide broad efficacy across CMC subtypes with limited host toxicity.

In this study using the spontaneous CMC model, we show that eCPMV intratumoral immunotherapy is safe and well tolerated; it induced a systemic response with significant transient changes in neutrophil counts and a modified immune response in the TME associated with an increase in neutrophils, dendritic cells, T cells and B cells. Importantly, the treatment caused variable tumor reduction in both the injected tumor and noninjected tumors in canine luminal and TN subtypes.

Regarding the safety of eCPMV immunotherapy, and in agreement with previous studies [24], hematology, biochemistry, and blood plasma assays did not show abnormal changes in any of the indicators used to track potential patients’ adverse events, and QOL was not affected in the treated dogs.

In regard to systemic immune changes, eCPMV induced fluctuations in blood lymphocytes and monocytes. A significant increase in the number of mature and immature blood neutrophils in most of the dogs was observed, with the levels remaining high after the second eCPMV injection and returning to basal levels by D45. A recent study in healthy dogs categorized CD14^+^ blood monocytes as MHCII^+^CD4^−^ (Mo1), MHCII^+^CD4^+^ (Mo2), and MHCII^−^CD4^+^ (Mo3) subtypes [37], and found significantly lower numbers of Mo2 and Mo3 subtypes than Mo1. Higher basal reactive oxygen species were produced in Mo2 and Mo3 than Mo1, implicating Mo2 and Mo3 subtypes in the promotion of inflammation and neoplastic progression [37]. In our study, the treatment generated higher numbers of Mo1 than Mo2 and Mo3 subtypes, with minor variations during the eCPMV treatment period. Of note, we observed a significant decrease in plasma MCP-1 levels after the first eCPMV injection, with an increase to basal levels after the second injection, and a steady decrease in IL-10. MCP-1 has chemotactic activity for monocytes and may be involved in monocyte recruitment [38], and IL-10 is associated with the inhibition of proinflammatory cytokines in monocytes [39]. However, whether they mediate different effects on each monocyte subtype remains unknown.

With regard to the local immune response in eCPMV-treated tumors, in contrast to the non-significant changes in the systemic Treg^+^/CD8^+^ ratio, eCPMV immunotherapy induced a significant decrease in the FoxP3^+^/CD3^+^ ratio in injected tumors as indicated via IHC. We also observed that eCPMV immunotherapy induced a significant increase in the number of innate immune cells in the injected tumor mass, with a striking six-fold increase in neutrophils numbers. These findings are in agreement with our previous studies demonstrating that eCPMV particles are rapidly taken up and they activate neutrophils in the TME as an important part of the antitumor immune response [21]. Further, we have previously demonstrated that the eCPMV-induced immune response in mice is relayed through TLR2 and 4 [26]. Within this context, it is worth noting that neutrophils express both TLR2 and TLR4 [40], secrete a variety of cytokines and chemokines that, among other functions, recruit macrophages, dendritic cells, T cells [41], and neutrophil-derived factors which drive B cell expansion and plasma cell differentiation [42]. Furthermore, neutrophils can influence T cells in various ways [43], including migrating to lymph nodes, and presenting antigens to T cells [44]. RNAseq demonstrated an increase in activated DCs, and a decrease in M1 macrophages in treated tumors. Hence, we postulate that eCPMV-activated neutrophils generate cytokines and chemokines to locally activate dendritic cells, macrophages, and T and B cells. Activated antigen-presenting cells may migrate to the lymph nodes, prime T cells and generate a systemic antitumor immune response responsible for tumor reduction in distant noninjected nodules, as observed in most of the CMC patients. We further noted differences in pathways related to extracellular fibers, which suggests a general reorganization of the TME upon eCPMV treatment.

Weekly eCPMV injection for two weeks resulted in tumor reduction in all injected tumors in all dogs, confirming the local efficacy of eCPMV immunotherapy observed in canine oral melanoma [24] and CIMC patients [25]. While reduction in the treated tumor is of great value, the goal of intratumoral immunotherapy is to also generate systemic antitumor immunity to protect against metastatic disease. Importantly, eCPMV immunotherapy also caused a reduction in benign tumors and dysplasias and invasive tumors of various histological grades in noninjected nodules in the ipsilateral and contralateral mammary glands of the treated dog. The ‘abscopal effect’, an immune-mediated response to radiation by tumor cells located distant from the irradiated site [45], implies elimination of tumor cells in the nontreated tumor sharing clonality with the treated/injected tumor. However, it has been demonstrated that human bilateral BC tumors [46,47] and multiple canine tumors in the same animal are generally not clonal, with independent somatic mutations and copy number alterations [48]. One mechanism that could explain systemic immune response against non-clonal breast tumors is an immune response against tumor-associated antigens, that, unlike neoantigens, are often common between nonclonal tumors of a given type. Future studies addressing the systemic effect of CPMV on distant tumors of different histopathology, and probably, genomic features in CMC patients are warranted.

As of July 2022, all eCPMV-treated dogs were doing well without any reported metastatic events. The observed tumor burden reduction in injected and noninjected tumors in both mammary chains suggests that the systemic immune response suppresses distant metastases. Although the study was not powered to evaluate the clinical effect of eCPMV immunotherapy on CMC patient outcomes (DFS and OS), a comparison with historical CMC cases treated at the same institution demonstrated that recurrences and distant metastases were observed during the two years follow-up in 3%, 16%, and 59% of dogs with histopathologic grade I, II, and III, respectively [28].

The repeated administration of eCPMV nanoparticles was expected to lead to a production of anti-drug antibodies (ADAs) [49] which could alter the eCPMV pharmacokinetic and pharmacodynamic properties and thereby reduce or improve the antitumor response [50], or cause an adverse immune reaction. Pretreatment detection of anti-CPMV antibodies in blood of all five eCPMV-treated dogs which were assayed indicates the common existence of circulating anti-CPMV antibodies. In the three patients with lower levels of anti-CPMV antibodies, the level of these antibodies increased into the same high range of the other two dogs within 14 days of first treatment (Appendix A). Subsequent changes in titers of anti-CPMV antibodies were stable for over 100 days. Prior studies detected anti-CPMV antibodies in more than half of human samples assayed [49], likely due to prior exposure to CPMV in food for both humans, and in this case, dogs. Despite initial detection of increased anti-CPMV titers following eCPMV immunotherapy in the three cases, no adverse reactions occurred, supporting the safety of this approach. Since a considerable tumor reduction was observed in both injected and noninjected tumors and benign masses, anti-CPMV antibodies do not block the efficacy of eCPMV immunotherapy. These findings agree with our results in a murine model of ovarian cancer where prior exposure to CPMV creating anti-CPMV IgG titers improved the efficacy of CPMV immunotherapy [35].

Although this open label prospective study demonstrated clear efficacy of eCPMV immunotherapy in CMC patients, the study conclusions are limited by low patient numbers and the lack of detailed molecular analysis of noninjected tumors. This study was focused on the injected tumor, and noninjected tumors were only used to track the eCPMV-induced systemic response by %TGI. In the future, and based on this experience, we will include those tumors for molecular studies.

In a previous study we demonstrated the efficacy of intratumoral eCPMV in CIMC, an aggressive, highly metastatic, and lethal form of mammary tumors in dogs [25]. Although surgery is not an option for CIMC patients, and most of the treatment is focused on palliative care and quality of life [51,52], eCPMV efficacy was sufficient to allow for surgery in two out of five CIMC patients, and the therapy was associated with improved survival. In the present study, we further demonstrated good efficacy in injected tumors, and the induction of a systemic response associated with tumor reduction in noninjected tumors in the ipsilateral and contralateral mammary chains. Although the study was not powered for survival analysis, the comparison with historical cases from the same institution highlights the absence of relapses/metastasis in eCPMV-treated patients. In addition, the RNAseq analysis provided an insight into the changes induced by eCPMV in cell cycle and cytoskeletal processes, as well as immune-related pathways in the TME. Future studies will focus on expanding the therapy to a larger number of CMC patients and evaluate combinations of eCMPV immunotherapy with chemotherapy and/or immune checkpoint inhibitors along with biomarker studies. Based on mouse studies, we expect that combining eCPMV immunotherapy with checkpoint inhibitors administered intratumorally or systemically will result in better efficacy than single-agent immune monotherapy [53,54].

## 5. Conclusions

Our findings indicate that eCPMV is a potent immunotherapy for treating canine mammary tumors; it appears to be safe and well tolerated, does not induce an exaggerated release of cytokines, or generate adverse immune reactions in treated dogs, and modulates immune cell populations. Furthermore, the treatment consistently shrank both treated and untreated tumors, demonstrating potential value for subsequent surgical treatment as well as for suppressing metastases. The studies support deploying eCPMV immunotherapy as a potential novel and effective neoadjuvant immunotherapy against a broad range of human BC subtypes.

## Figures and Tables

**Figure 1 cells-12-02241-f001:**
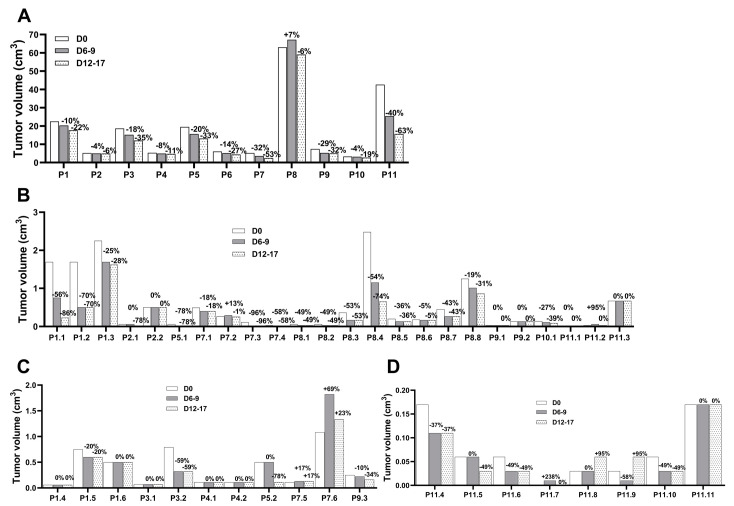
eCPMV immunotherapy induced tumor regression in target injected and noninjected CMC. (**A**) CMC patients received the first eCPMV injection in the target tumor at D0 and the second injection at D6–9, followed by surgery at D12–17. Tumor reduction was observed in noninjected tumors in both ipsilateral (**B**) and contralateral (**C**,**D**) mammary chains. Tumor volume is on the *y*-axis and the CMC patients with the tumor number, as described in Table 3, are listed in the *x*-axis. The percentage of tumor growth inhibition (%TGI) at the different days relative to D0 is indicated on top of each column.

**Figure 2 cells-12-02241-f002:**
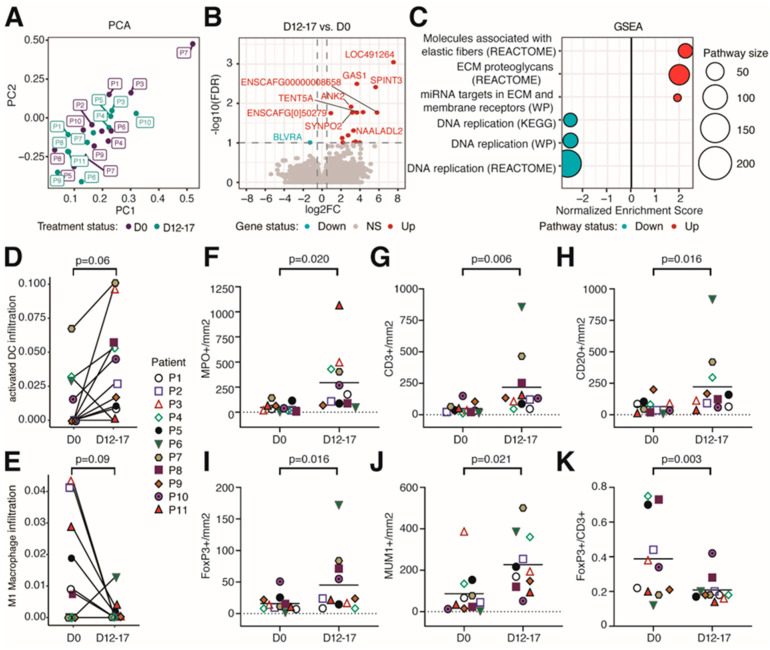
eCPMV immunotherapy modulated the tumor microenvironment in treated tumors. (**A**) Principal component analysis (PCA) of all samples analyzed via RNA-seq. (**B**) Volcano plot of differentially expressed genes comparing D12–17 to D0 gene expression. The horizontal dotted line indicates FDR = 0.1. Vertical dotted lines indicate a log_2_FC of −0.5 and 0.5. (**C**) Selected significant pathways from Gene Set Enrichment Analysis (GSEA) results comparing gene expression between D0 and D12–17 samples. (**D**) Infiltration of activated dendritic cells comparing D0 and D12–17 samples. *p*-value calculated using paired Wilcoxon signed-rank test. (**E**) Infiltration of M1 macrophages between D0 and D12–17. *p*-value calculated via paired Wilcoxon signed-rank test. Through IHC, canine patients treated with eCPMV immunotherapy had a significant increase in the number of neutrophils (MPO+; (**F**)), T lymphocytes (CD3+; (**G**)), B lymphocytes (CD20+; (**H**)), Treg lymphocytes (FoxP3+; (**I**)), and plasma cells (MUM1+; (**J**)), and the FoxP3+/CD3+ ratio (**K**) was significantly reduced. Each companion dog is represented by a colored individual shape as indicated in plot 3D-E. *X*-axis indicates the day when biopsies were taken; *y*-axis indicates the number of cells per square millimeter. *p*-value obtained via paired Student *t*-test.

**Table 1 cells-12-02241-t001:** Epidemiological and clinicopathological characteristics of eCPMV-treated CMC patients.

Patient	Age, y.	Weight, kg	Clin. Stage	Histotype; Carcinoma Grade	LNI	* TS, cm	** TS, cm^3^	Molecular Subtype	Adj. Ther.	Rec. Met.	OS, Days	Status
P1	10.4	39.2	II	I	No	5.0	22.5	Luminal A	No	No	1345	Alive
P2	5.8	37.8	I	I	No	2.6	5.2	Luminal A	No	No	1226	Alive
P3	9.1	20.5	IV	I	Yes	6.2	18.6	Luminal A	No	No	1250	Alive
P4	8.4	6.5	II	I	No	3.3	5.3	Luminal A	No	No	1202	Alive
P5	8.0	32.7	II	I	No	4.2	19.5	Luminal A	No	No	1135	Alive
P6	9.6	33.2	IV	III	Yes	6.6	6.0	TN	No	No	217	NCRD
P7	13.0	9.2	I	I	No	2.9	5.2	Luminal A	No	No	411	Alive
P8	12.1	30.3	III	I	No	7.5	63.0	Luminal A	No	No	425	Alive
P9	10.8	9.3	I	I	No	2.7	7.5	Luminal A	No	No	449	Alive
P10	12.5	23.3	I	III	No	2.3	3.3	TN	mCTX + F	No	414	Alive
P11	14.8	6.1	III	II	No	5.9	42.6	Luminal B	mCTX	No	267	Alive

Legends: Age at diagnosis in years; TN, triple-negative; * TS, tumor size, refers to the largest diameter of the target tumor in cm, and ** tumor volume in cm^3^; Clin. Stage, clinical stage; LNI, regional lymph involvement; Adj. Ther., adjuvant therapy; mCTX, metronomic Cyclophosphamide; F, Firocoxib; Rec. met., recurrence/metastasis; OS, overall survival time; NCRD, non-cancer-related death.

**Table 2 cells-12-02241-t002:** Tumor volume changes in eCPMV-injected tumors in CMC patients.

Patient	Day	Tv, cm^3^	%TGI	*p* Value
P1	D0	22.5		
(L2)	D8	20.3	−10.0	
	D16	17.6	−21.6	0.037
P2	D0	5.2		
(R5)	D7	5.0	−3.8	
	D14	4.9	−5.8	0.121
P3	D0	18.6		
(R3)	D7	15.2	−18.4	
	D13	12.0	−35.4	0.011
P4	D0	5.3		
(R4)	D6	4.9	−8.1	
	D13	4.8	−10.8	0.212
P5	D0	19.5		
(R4)	D9	15.6	−20.1	
	D17	13.2	−32.5	0.087
P6	D0	6.0		
(L5)	D6	5.2	−14.3	
	D13	4.4	−27.0	<0.001
P7	D0	5.2		
(L5)	D8	3.6	−31.5	
	D15	2.5	−53.2	0.068
P8	D0	63.0		
(L4)	D7	67.2	6.6	
	D12	59.0	−6.3	0.642
P9	D0	7.5		
(L5)	D7	5.3	−29.0	
	D13	5.1	−32.0	0.238
P10	D0	3.3		
(L3)	D6	3.2	−4.3	
	D13	2.7	−19.1	0.213
P11	D0	42.6		
(R3)	D8	25.4	−40.3	
	D15	15.7	−63.2	0.073

Legends: The parenthesis under the patient is referred to the tumor location: L: left chain; R: right chain; the numbers indicate the location in the mammary chain from cranial to caudal; D0 refers to the day when eCPMV immunotherapy started; D with a number indicate the day measurements were taken; Tv, tumor volume; %TGI, percentage of tumor growth inhibition; *p* values obtained via regression analysis from D0 to D14 as described in Methods.

**Table 3 cells-12-02241-t003:** TGI changes in noninjected tumors in eCPMV-treated dogs.

Ipsilateral Chain																		
ID	P1—Gr. I		P2—Gr. I		P5—Gr. I		P7—Gr. I		P8—Gr. I				P9—Gr. I		P10—Gr. III		P11—Gr. II	
	V, cm^3^	%TGI	V, cm^3^	%TGI	V, cm^3^	%TGI	V, cm^3^	%TGI	V, cm^3^	%TGI	V, cm^3^	%TGI	V, cm^3^	%TGI	V, cm^3^	%TGI	V, cm^3^	%TGI
	P1.1—Gr. I		P2.1—Benign		P5.1 Gr. I		P7.1—Gr. II		P8.1—Gr. I		P8.5—Benign		P9.1—HD		P10.1—Gr. I		P11.1—Gr. I	
D0	1.69		0.06		0.06		0.49		0.06		0.20		0.03		0.14		0.01	
DTx2	0.75	−55.56	0.06	0.00	0.01	−78.40	0.40	−18.24	0.03	−48.80	0.13	−36.36	0.03	0.00	0.11	−26.48	0.01	0.00
DSx	0.23	−86.11	0.01	−78.40	0.01	−78.40	0.40	−18.24	0.03	−48.80	0.13	−36.36	0.03	0.00	0.09	−39.24	0.01	0.00
	P1.2—HD		P2.2—Benign				P7.2—Gr. I		P8.2—Gr. I		P8.6—Gr. I		P9.2—Benign				P11.2—Gr. II	
D0	1.69		0.50				0.26		0.06		0.18		0.13				0.03	
DTx2	0.50	−70.37	0.50	0.00			0.29	12.50	0.03	−48.80	0.17	−4.72	0.13	0.00			0.06	95.31
DSx	0.50	−70.37	0.50	0.00			0.25	−1.12	0.03	−48.80	0.17	−4.72	0.13	0.00			0.03	0.00
	P1.3—HD						P7.3—HD		P8.3—Gr. I		P8.7—Gr. I						P11.3—Gr. II	
D0	2.25						0.11		0.36		0.45						0.67	
DTx2	1.69	−25.00					0.004	−96.30	0.17	−52.95	0.26	−42.54					0.67	0.00
DSx	1.62	−27.96					0.004	−96.30	0.17	−52.95	0.26	−42.54					0.67	0.00
							P7.4—Benign		P8.4—Benign.		P8.8—Gr. I							
D0							0.03		2.48		1.25							
DTx2							0.01	−57.81	1.15	−53.56	1.01	−19.36						
DSx							0.01	−57.81	0.65	−73.74	0.86	−31.23						
**Contralateral chain**																		
**ID**	**P1—Gr. I**		**P3—Gr. I**		**P4—Gr. I**		**P5—Gr. I**		**P7—Gr. I**		**P9—Gr. I**		**P11—Gr. II**					
	**V, cm^3^**	**%TGI**	**V, cm^3^**	**%TGI**	**V, cm^3^**	**%TGI**	**V, cm^3^**	**%TGI**	**V, cm^3^**	**%TGI**	**V, cm^3^**	**%TGI**	**V, cm^3^**	**%TGI**	**V, cm^3^**		**%TGI**	
	P1.4—Benign		P3.1—Benign		P4.1—Benign		P5.2—Unknown		P7.5—Gr. I		P9.3—Benign		P11.4—Unknown		P11.8—Unknown			
D0	0.06		0.07		0.11		0.50		0.11		0.25		0.17		0.03			
DTx2	0.06	0.00	0.07	0.00	0.11	0.00	0.50	0.00	0.13	16.67	0.22	−10.00	0.11	−37.03	0.03		0.00	
DSx	0.06	0.00	0.07	0.00	0.11	0.00	0.11	−78.40	0.13	16.67	0.16	−33.88	0.11	−37.03	0.06		95.31	
	P1.5—HD		P3.2—Benign		P4.2—Benign				P7.6—Gr. I				P11.5—Unknown		P11.9—Unknown			
D0	0.75		0.79		0.11				1.08				0.06		0.03			
DTx2	0.60	−20.00	0.32	−59.17	0.11	0.00			1.82	68.75			0.06	0.00	0.01		−57.81	
DSx	0.60	−20.00	0.32	−59.17	0.11	0.00			1.33	22.97			0.03	−48.80	0.06		95.31	
	P1.6—Gr. I												P11.6—Unknown		P11.10—Unknown			
D0	0.50												0.06		0.06			
DTx2	0.50	0.00											0.03	−48.80	0.03		−48.80	
DSx	0.50	0.00											0.03	−48.80	0.03		−48.80	
													P11.7—Unknown		P11.11—Unknown			
D0													0.004		0.17			
DTx2													0.014	237.50	0.17		0.00	
DSx													0.004	0.00	0.17		0.00	

Legend: %TGI at D0 indicates the %TGI of eCPMV-injected tumor used as a reference; the histopathologic Gr. of the injected tumor is indicated on top of the table, and of each individual nodule along with its location in the mammary chain; V, tumor volume; P1.1, P1.2, etc., location of the untreated tumor in the mammary chain; cm^3^; cubic centimeters; Gr., grade; HD, hyperplasia-dysplasia. D0, day of first eCPMV injection; DTx2 day of second injection; DSx, day of surgery.

## Data Availability

The data presented in this study are openly available in the NCBI Gene Expression Omnibus under GEO accession number GSE242689.

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
