# Peer review of "Neoadjuvant Intratumoral Immunotherapy with Cowpea Mosaic Virus Induces Local and Systemic Antitumor Efficacy in Canine Mammary Cancer Patients"

_cells, 2023, doi:10.3390/cells12182241_

Round 1

Reviewer 1 Report

The authors have evaluated anticancer efficacy of cowpea mosaic virus-liked particles (eCPMV) in canine mammary cancer models.

Tumor reduction was observed in both injected and non-injected tumors. Immune cell population analysis from blood and tumors indicated increased infiltration of anticancer immune cells. 

Comments:

1) Line#32,cowpea mosaic virus or cowpea mosaic virus-liked particles?

2) What x-axis (P1.1, P1.2) indicates  in figure 1B, 1C, and 1D? Does it indicate  tumor numbers?

3) Line#96, as per results, Tregs lymphocytes were decreased. Please check. 

4) Figure S10, as per the figure, FoxP3+ cells were decreased, please check.

Author Response

Reviewer 1.

The authors have evaluated anticancer efficacy of cowpea mosaic virus-liked particles (eCPMV) in canine mammary cancer models.

Tumor reduction was observed in both injected and non-injected tumors. Immune cell population analysis from blood and tumors indicated increased infiltration of anticancer immune cells. 

Comments:

1) Line#32,cowpea mosaic virus or cowpea mosaic virus-liked particles?.

Answer: It should be virus-like particles. The text in line 32 now states: ‘virus-like particles’.

2) What x-axis (P1.1, P1.2) indicates  in figure 1B, 1C, and 1D? Does it indicate  tumor numbers?

Answer: We appreciate this request to clarify the statement. The paragraph in lines 210-216, Legend to Figure 1, was modified to accommodate this suggestion. Line’s 214 text was changed to “Tumor volume is on the y-axis and the CMC patients with the tumor number as described in Table 3 are listed in the x-axis.”

3) Line#96, as per results, Tregs lymphocytes were decreased. Please check. 

Answer: We greatly appreciate this remark which goes along with the next question.

4) Figure S10, as per the figure, FoxP3+ cells were decreased, please check.

Answer: As indicated in line 96 ‘A significant increase in the numbers of neutrophils, T lymphocytes, B lymphocytes, Treg lymphocytes, and plasma cells was observed….” The statement is correct:The significant increase in Tregs is illustrated in figure 2I, and the quantitation is presented in table S11 (15.9 vs. 45.3 cells/mm2; 2.8-fold increase; p= 0.016). However, as the reviewer pointed out, we failed with the immunohistochemistry illustration. Hence, to correct this awful mistake from our side, Figure S10 was modified to add a correct image for the Tregs immunostaining.

We also realized that the sentence in line 110 “and MUM1+ plasma cells were significantly increased (P=0.021, 2J)” is redundanta because these are plasma cells described already within first sentence (line 97). So, to correct this redundancy, we deleted that sentence and added the location of plasma (MUM1+ cells) in the second sentence. The paragraph (lines 95-105) now reads:

Additional changes in immune cell infiltration were detected by IHC analysis. A significant increase in the numbers of neutrophils, T lymphocytes, B lymphocytes, Treg lymphocytes,  and plasma cells was observed in posttreatment tumor biopsies when compared to pretreatment tumor biopsies (P<0.02 for all; figure 2F, G, H, J, figure S10, and table S11). Although a significant increase in the number of Treg lymphocytes was observed (P=0.016; figure 2I), the Treg/Total T cell ratio (FoxP3+/CD3+) significantly decreased (P=0.003; figure 2K). eCPMV immunotherapy did not induce significant changes in Ki67 proliferation index (table S11). These findings highlight the strong immunogenicity of eCPMV nanoparticles which induced a significant increase of innate and adaptive immune cell populations within the TME in eCPMV-treated CMC patients.

While this comment does not affect the review, it should be noted that the numbering in both the pdf and the original doc we received changed starting on page 9 where the numbering starts again with 1.

Reviewer 2 Report

Even at the very beginning of the publication under the title " Neoadjuvant Intratumoral Immunotherapy with Cowpea Mosaic Virus Induces Local and Systemic Antitumor Efficacy in Canine Mammary Cancer Patients" is quite impressive with the list of authors of the publication as well as the scientific centers cooperating in the creation of this work. In my humble opinion, it is the international cooperation of specialists in many different fields that makes it possible to solve increasingly difficult problems in human and selected animal medicine. On the other hand, the deactivation of the host's immune system by the developing cancerous tumor is one of the main problems in the fight by the organism attacked by the development of a cancerous tumor Hence, the treatment through immuotherapy using appropriately prepared viruses is very interesting. 

Review of the literature made in a thorough and clear way presented to the reader. Objectives and research hypotheses presented in a brief but understandable manner. Materials and research methods well described and understandable allowing their use by other researchers . Very interesting results described in detail using correct statistics. Discussion conducted in a factual manner and critical of one's own results when confronted with the results of other authors . The authors also give the weaknesses of their publication and suggestions for further research directions in this promising field . Basically, this description represents my point of view on this publication but the lack of critical comments may be due to my insufficient knowledge in this area of science or to the really good work of all the authors mentioned. 

Author Response

Revier 2.

Even at the very beginning of the publication under the title " Neoadjuvant Intratumoral Immunotherapy with Cowpea Mosaic Virus Induces Local and Systemic Antitumor Efficacy in Canine Mammary Cancer Patients" is quite impressive with the list of authors of the publication as well as the scientific centers cooperating in the creation of this work. In my humble opinion, it is the international cooperation of specialists in many different fields that makes it possible to solve increasingly difficult problems in human and selected animal medicine. On the other hand, the deactivation of the host's immune system by the developing cancerous tumor is one of the main problems in the fight by the organism attacked by the development of a cancerous tumor Hence, the treatment through immuotherapy using appropriately prepared viruses is very interesting. 

Review of the literature made in a thorough and clear way presented to the reader. Objectives and research hypotheses presented in a brief but understandable manner. Materials and research methods well described and understandable allowing their use by other researchers . Very interesting results described in detail using correct statistics. Discussion conducted in a factual manner and critical of one's own results when confronted with the results of other authors . The authors also give the weaknesses of their publication and suggestions for further research directions in this promising field . Basically, this description represents my point of view on this publication but the lack of critical comments may be due to my insufficient knowledge in this area of science or to the really good work of all the authors mentioned. 

Answer: We do appreciate the kind words and recognition that efforts like this demand international collaboration to get the best of each team to obtain clinical valuable information for both our companion dogs and, potentially, human breast cancer patients. The number of dogs is not what we would like to have, but this study strongly suggests that extending the work to a large number of dogs and more genomic data will be relevant to both species and worth the investment.